# Broadband Terahertz Photonic Integrated Circuit with Integrated Active Photonic Devices

**Amlan Kusum Mukherjee *** , **Mingjun Xiang** and **Sascha Preu**

Department of Electrical Engineering and Information Technology, Technical University of Darmstadt, Merckstr. 25, 64283 Darmstadt, Germany; xiang@fias.uni-frankfurt.de (M.X.); sascha.preu@tu-darmstadt.de (S.P.)
* Correspondence: amlan.mukherjee@tu-darmstadt.de

**Abstract:** Present-day photonic terahertz (100 GHz–10 THz) systems offer dynamic ranges beyond 100 dB and frequency coverage beyond 4 THz. They yet predominantly employ free-space Terahertz propagation, lacking integration depth and miniaturisation capabilities without sacrificing their extreme frequency coverage. In this work, we present a high resistivity silicon-on-insulator-based multimodal waveguide topology including active components (e.g., THz receivers) as well as passive components (couplers/splitters, bends, resonators) investigated over a frequency range of 0.5–1.6 THz. The waveguides have a single mode bandwidth between 0.5–0.75 THz; however, above 1 THz, these waveguides can be operated in the overmoded regime offering lower loss than commonly implemented hollow metal waveguides, operated in the fundamental mode. Supported by quartz and polyethylene substrates, the platform for Terahertz photonic integrated circuits (Tera-PICs) is mechanically stable and easily integrable. Additionally, we demonstrate several key components for Tera-PICs: low loss bends with radii ~2 mm, a Vivaldi antenna-based efficient near-field coupling to active devices, a 3-dB splitter and a filter based on a whispering gallery mode resonator.

**Keywords:** dielectric waveguides; resonator filters; terahertz; terahertz system-on-chip; Vivaldi antennas





## 1. Introduction

In the past few decades, huge advances have been made in creating sources and detectors working to bridge the so-called THz gap (0.1–10 THz). Currently available table-top, non-cryogenic THz sources can generate up to tens of milliwatts of power at the lower end of the THz range [1–4]. Combined with highly sensitive THz receivers, both photonic and electronic THz systems provide dynamic ranges of 120–140 dB [5,6]. Similar to integrated circuits in the RF and microwave domain [7,8] or photonic integrated circuits at the telecom bands in the near infrared there exist integrated terahertz systems. These include electronic systems based on fundamental oscillators at the lower end of the terahertz range [9] or frequency multiplier-based systems [6], both frequently using hollow metallic waveguides with microstrip transitions to active elements, as well as dielectric waveguide-based systems [10–12]. While the waveguide concepts generally allow for large bandwidths, the lack of broadband characterization tools as well as comparatively narrowband transitions to active elements typically restrict the operational bandwidth to typically not more than 10–50% of the centre frequency.

To date, the most powerful systems in terms of dynamic range are electronic systems based on hollow metal waveguides, at least below ~1 THz. Their success is based on the comparatively low loss as compared to coplanar waveguides and microstrip circuitry and well engineered transitions to active elements with typical losses lower than 1 dB [13]. These transitions are necessary as semiconductor-based active components such as Schottky diodes still have to be fabricated in a microstrip circuitry or similar [14]. The down side of these transitions is the requirement of strongly wavelength-dependent (mode) field matching techniques, such as positioning the transition at a local field maximum, and

impedance matching techniques, such as implementation of a sequence of $\lambda/4$ layers or stubs as well as resonant filters. The strong wavelength dependence of these is the main reason for the bandwidth limitation. At frequencies beyond 1 THz, further losses of metal-based waveguides appear due to surface roughness, the Skin effect as well as alignment errors and imperfect transitions. Still, electronic systems are the systems of choice for circuit analysis (e.g., vector network analysis) in the lower Terahertz range. The transitions to the circuits, such as GSG probes, are typically manufactured on rigid ceramics in order to minimize mechanical failure and wear off.

Dielectric waveguides do not show conductive losses. However, dielectric waveguides also face challenges: the lower cut-off of higher order modes may cause radiation and mode conversion losses at bends which may lead to dysfunctional components such as transitions to active devices, similar to their electronic counterparts. Furthermore, the waveguide losses, mode field diameter and the wave impedance are strongly frequency-dependent. Despite these challenges, several groups successfully demonstrated dielectric waveguide integrated terahertz systems in the past decade. Han et al. and Chen et al. [15,16] demonstrated polymer-based waveguides with operational bandwidths beyond 1 THz and losses of around 3–4 dB/cm; however, without any waveguide-based components. SiGe monolithic microwave integrated circuits (MMICs) working between 110–170 GHz were demonstrated in ref. [17]. Headland et al. showed silicon waveguide-based THz systems working between 260–390 GHz [18]. Multiple topologies of silicon on insulator (SoI)-based waveguides [19,20] and integrated Terahertz resonators [21,22], waveguides based on silicon-BCB-quartz platform [12] and suspended silicon THz waveguides [23,24] report relatively low losses between 0.4–1 dB/cm, and feature a bandwidth ranging between 100–250 GHz. Free-standing multimode dielectric waveguides with excellent transitions were demonstrated using dielectric rod waveguides [25,26]. Alternatively, photonic crystal-based silicon waveguides [27–30] and its effective medium-based variants [31] have also demonstrated to feature lower losses of 0.05–0.1 dB/cm at bandwidths from 80–150 GHz to around 300 GHz. For comparison, rectangular hollow metallic waveguides feature losses around 1.3–1.9 dB/cm at 1 THz and similar relative bandwidth [32].

Most of the aforementioned dielectric waveguide structures were operated with comparatively narrowband electronic systems, simply because of their availability and high dynamic range. In the past, photonic terahertz systems with a frequency coverage of several octaves [33–35] became competitive in terms of dynamic range, particularly above 1 THz. Their large operational bandwidth is achieved by mixing frequency components of a laser signal (typically 800 nm or 1550 nm telecom lasers) in a semiconductor device. These frequency components either result using a femtosecond laser pulse with >1 THz bandwidth or by mixing two (usually continuous-wave) lasers differing by the desired THz frequency (e.g., a 1550 nm laser and a 1558 nm laser). For the latter, the tuning happens in the optical domain with relative ease: 1 THz corresponds to 0.5% tuning at 1550 nm which is straight forward for many laser concepts, yet enabling systems with tuning range beyond 3.5 THz [36]. As opposed to their electronic-integrated counterparts, photonic THz systems usually use bulky free space optics such as parabolic mirrors or lenses with little dependence of the propagation loss on frequency. To the knowledge of the authors, there are no photonic systems with on-chip circuitry measurement capability, such as vector network analyzers, that excel the operational bandwidth of electronic ones.

In this work, we present a platform for this purpose based on mechanically rigid SoI THz waveguides and characterize several key components for photonic integrated circuits [37] over a bandwidth of 0.5–1.6 THz, about a factor of three larger than previous investigations. The waveguides are manufactured from highly resistive float zone silicon (HRFZ-Si, $\epsilon_{r,Si} = 11.678 \pm 0.001$, $\sigma_{si} < 0.01$ S/m) supported by either crystalline quartz ($\epsilon_{r,qz} = 4.52$) [38] or high-density polyethylene (HDPE, $\epsilon_{r,PE} = 2.4$) substrates for ease of integration. These materials, along with other polymers such as Cyclo-Olefin Copolymer or Teflon, are known to have low refractive indices, accompanied by very low loss up to $\sim$2 THz or even higher. Low refractive index of the waveguide substrate is essential to

have an appreciable leakage-free bandwidth. We also show transitions to active elements integrated in a planar dielectric waveguide. The same transitions may be exploited as on-chip GS connections to THz circuits in the future. The support by rigid substrates ascertains mechanical rigidity and stability.

## 2. Waveguide Architecture

The waveguide architecture consists of a rectangular HRFZ-Si waveguide mounted on a low loss, low refractive index substrate as illustrated in Figure 1a). Dielectric waveguides theoretically offer infinite single-mode bandwidth; however a non-homogeneous surrounding media limits its lowest operating frequency [39]. The field of a selected mode experiences not only the refractive index of the rectangular waveguide core $n_{WG}$ but also that of the surrounding media due to the evanescent field components of the mode. This results in an effective refractive index, composed of an integral of the local refractive indices, weighted by the local power across the mode profile. Unlike metallic waveguides, the non-homogeneous waveguide topology as shown in Figure 1a,b, where air ($n_{air} \approx 1$) and a dielectric medium ($n_{diel}$) is situated above and below the waveguide, respectively, ($n_{air} < n_{diel} < n_{WG}$), a lower cut-off frequency is introduced when the evanescent field in the air is sufficiently large to cause the effective refractive index to drop below the value of the substrate $n_{diel}$. This causes the wave to leak into the substrate and is not guided further. Figure 1c) shows the simulated propagation constant ($\beta$) for the propagating modes for such a designed waveguide. At $f \leq 0.5$ THz, propagation constant of the fundamental mode $HE_{11}^x$ is lower than that of bulk quartz, causing a part of the wave to leak towards the quartz substrate. The simulation snapshots in the inset concur with this behaviour. A polyethylene substrate ($\epsilon_{r,PE} = 2.4$) reduces the lower 3-dB cut-off frequency to 0.411 THz. As frequency increases, the wave becomes strongly confined within the HRFZ-Si structure. Hence, the substrate leakage decreases and, as a result, the transmission losses depend only on material properties.

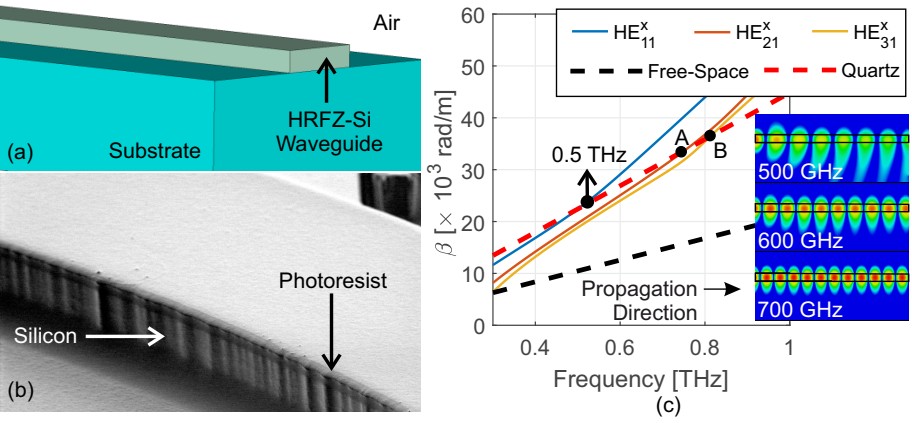

**Figure 1.** (**a**) Topology of the designed dielectric waveguides. (**b**) Microscopic image of an etched waveguide. The photoresist mask is visible on top of the waveguides. Side-wall irregularities are caused due to the pixelated nature of the lithographic mask. (**c**) Simulated values of the propagation constant ($\beta$) of the first 3 modes. Points A (0.75 THz) and B (0.81 THz) show the lower cut-off frequencies for $HE_{21}^x$ and $HE_{31}^x$ modes, respectively. Simulation snapshots in the inset shows waves leaking into the substrate for $f_{THz} \leq 0.5$ THz. The black rectangular outline represents the HRFZ-Si waveguide. The dielectric below is quartz ($n = 2.13$), the dielectric above is air ($n = 1$).

Figure 1c also shows that higher order modes $HE_{21}^x$ and $HE_{31}^x$ start propagating above 0.75 THz and 0.81 THz, respectively. Thus, like its metallic counterpart, a dielectric waveguide operating between 0.5–2 THz supports multiple modes at the higher frequency end. We show in the upcoming sections that with appropriate bend designs and controlled excitation, the dielectric waveguides can be practically operated in its fundamental mode even above its 3-dB dispersion bandwidth of 0.5–0.75 THz, without loss of functionality of

integrated components such as splitters/combiners and filters. The presented waveguides are planar and hence, any modal perturbation in the horizontal dimension can be controlled by lateral structural modifications, which becomes much more difficult for the vertical dimension with planar processing techniques. Thus, we designed the waveguide to support multiple modes along its width, but only a single mode along its height up to a frequency of 2 THz, resulting in the rectangular structure depicted in Figure 1a). The waveguides characterised in this work are 200 μm wide and 50 μm tall.

## 3. Fabrication and Measurement Setup

Rectangular waveguide structures are etched out of a 50 μm thick HRFZ-Si wafer, glued on a 525 μm thick quartz 2 inch quarter-wafer with a thin layer of superglue. The waveguides are fabricated either by deep reactive ion etching (DRIE) of a lithographically defined hard mask (hard-baked AZ® nLof 2035) on the silicon wafer or by laser ablation. DRIE employs a Bosch Process, which involves repetitive deposition and etching cycles using inductively coupled plasmas of $C_4F_8$ and $SF_6$. While processing, the silicon wafer is cooled to $-10$ °C from below, which necessitates the substrate to have high thermal conductivity. For laser ablation, the substrate material must feature a high melting point. Hence, both the processes were conducted using crystalline quartz as substrate. The main difference between waveguides produced by DRIE and laser ablation is the roughness of their side walls. In DRIE, the quality of the photolithographic layer determines the side-wall roughness, which is a few hundreds of nanometres for our case, whereas in ablation processes, wall roughness can be reduced to as low as tens of nanometres. Additionally, it is more difficult to maintain the stability of process parameters in DRIE than in laser ablation. However, DRIE processing of a 50 μm silicon quarter-wafer takes only 20–25 min to complete, and is chosen as the preferred method of fabrication. In comparison, laser ablation of a similar silicon wafer requires ∼7 h. After processing, some waveguide structures were transferred onto a HDPE substrate with lower dielectric constant and the former quartz substrate was removed.

Figure 2a shows the schematic of the setup. It consists of a Tx module with a p-i-n diode-based THz transmitter (Fraunhofer Heinrich Hertz Institute/Toptica photonics) followed by a parabolic mirror in order to achieve a collimated terahertz beam. A TPX (Polymethylpentene) lens (L1) couples the signal to the waveguide that features a 90° bend in order to exclude line-of-sight coupling. A second TPX lens (L2) couples the power to a Rx module, consisting of another parabolic mirror and a silicon lens-coupled ErAs:InGaAs photoconductive THz receiver with logarithmic-periodic antenna ($Rx_{LP}$) developed in-house [40]. The transmission coefficient is calculated by referencing to a measurement without waveguide and Tx and $Rx_{LP}$ modules facing each other with both TPX lenses in between.

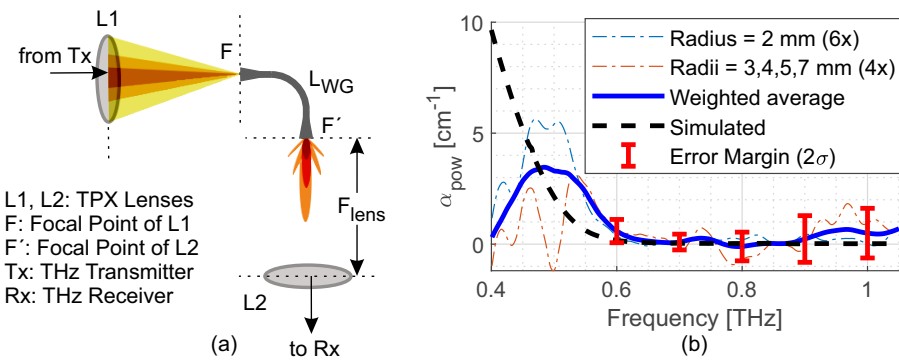

**Figure 2.** (**a**) shows the setup used for measuring the transmission through the waveguide structures. The Rx module is rotated by 90° in order to prevent line-of-sight coupling from Tx. All waveguide structures therefore include a 90° bend. (**b**) Measured power attenuation coefficient for the straight waveguide sections and its comparison with the simulated loss. $\alpha_{pow}$ is averaged over 25 GHz.

## 4. Waveguide Performance

### 4.1. Transmission Losses through Straight Sections

The total transmission loss is composed of two categories, namely losses proportional to the length of waveguides occurring at the straight sections and losses originating at waveguide bends or discontinuities due to radiation and mode conversion. To estimate losses of the first kind, losses introduced by the bends are removed by comparing waveguides of different lengths but subsets of same radius of curvature (ROC), similar to the cut back technique used for optical fibres, i.e., for two waveguides with same ROC but of lengths $l_1$ and $l_2$, the power attenuation coefficient ($\alpha_{pow}$) is calculated as:

$$\alpha_{pow} = -\frac{2}{l_1 - l_2} \ln\left(\frac{I_1}{I_2}\right), \tag{1}$$

where $I_1$ and $I_2$ are the measured current at the receiver, which are proportional to the incident terahertz field. Several waveguides with same bend radius but different lengths $l_{WG}$ between 15–35 mm are characterized. Furthermore, different waveguide bend radii between 1–11 mm were implemented with varying lengths of the straight sections. Figure 2b depicts the calculated average power attenuation coefficient with excellent agreement with the simulated values. Only below 0.47 THz, i.e., below the lower cut-off frequency where the waveguides lost their guiding properties, there is a noteworthy discrepancy. The transmission losses decrease from ≈8 dB/cm at 0.5 THz to ≈0.4 dB/cm at 0.65 THz. Above 0.9 THz, standing waves in the waveguides affect the data quality. A further plausible origin of these oscillations is inter-mode coupling between the fundamental and higher order modes. We show in simulations in Section 4 of the supplemental information that the power in the higher order modes is suppressed by at least 15 dB in the frequency range of interest. Furthermore, mode-inter-coupling and standing waves can be discriminated by investigating the time-domain traces of the measured data, as illustrated in Section 5 of the supplementary material. We therefore conclude that standing waves are the dominant origin of the oscillations. The measured absorption coefficient is larger than the simulated value of a HRFZ-Si waveguide placed on crystalline quartz, i.e., ≈0.13 dB/cm, due to the presence of a 5–9 µm layer of ethyl cyanoacrylate-based superglue ($\epsilon_r = 2.72$, $\tan\delta = 0.067$ at 1 THz [41]) underneath the waveguides. Excess loss due to superglue, though not detrimental to the waveguide operation, can be easily avoided using other low-loss adhesives, such as BCB [12].

Dielectric waveguides are prone to dispersion as the effective mode permittivity of the guided mode increases with increasing frequency as a larger fraction of the wave is concentrated within the high index HRFZ-Si guide. The dispersion depends on the group velocity of the travelling wave [42]. The dispersion factor $D$ is given by [43]

$$D = \frac{2\pi c_0}{\lambda^2} \cdot \frac{\partial^2 \beta}{\partial \omega^2} = \frac{f_{THz}^2}{2\pi c_0} \cdot \frac{d^2 \beta}{d f_{THz}^2}. \tag{2}$$

where $\beta = 2\pi f_{THz}\sqrt{\epsilon_r^{eff}(f_{THz})}/c_0$ is the propagation constant of the guided mode, $\omega$ is the angular frequency and $c_0$ is the speed of light in vacuum. Total dispersion for a HRFZ-Si waveguide of length $l_{WG}$ and the setup can be calculated from the phase of the detected THz field,

$$\Phi_{WG} = \beta \cdot l_{WG} + \beta_0 l_{FS} + \phi_{ant}(f_{THz}) + \phi_0, \tag{3}$$

where $l_{FS}$ is distance travelled by the THz wave in free-space with a propagation constant $\beta_0 = 2\pi f_{THz}/c_0$, $\phi_{ant}$ is an additional phase induced by dispersive emitting and receiving antennas [44] and $\phi_0$ is a constant. Similarly, the phase of the reference signal ($\Phi_{ref}$) is

$$\Phi_{ref} = \beta_0 l'_{FS} + \phi_{ant}(f_{THz}) + \phi'_0, \tag{4}$$

where $\phi_0'$ is another arbitrary constant and the free space propagation length, $l_{FS}'$ may differ from that of Equation (3). By subtracting the reference phase, $\Phi_{ref}$, from the phase with waveguide, $\Phi_{WG}$, removes the dispersion caused by the antennas, yielding

$$D = \frac{f_{THz}^2}{2\pi c_0} \cdot \frac{1}{l_{WG}} \frac{\partial^2}{\partial f_{THz}^2} \left( \Phi_{WG} - \Phi_{ref} \right). \tag{5}$$

Figure 3a,b show the measured average dispersion of the waveguides on quartz and HDPE substrates, respectively. The measured dispersion is of the same order of magnitude and shows the same frequency-dependent behaviour as the simulated ones for both cases, but feature strong standing wave patterns. We remark that the standing waves are present over the whole investigated frequency range and therefore cannot originate from mode beating as higher order modes cannot propagate at the lower end of the investigated frequency range. Chromatic dispersion of the presented waveguides limits their applicability in pulsed-THz setups. However, dispersion compensation techniques can be further employed, e.g., by etching slits into the waveguide (slotted waveguides) [45] or by introducing photonic-crystal structures [46], for extending their applicability in THz pulsed systems. In CW systems, dispersion effects can easily be removed by normalization to a reference measurement without any influence on the data quality.

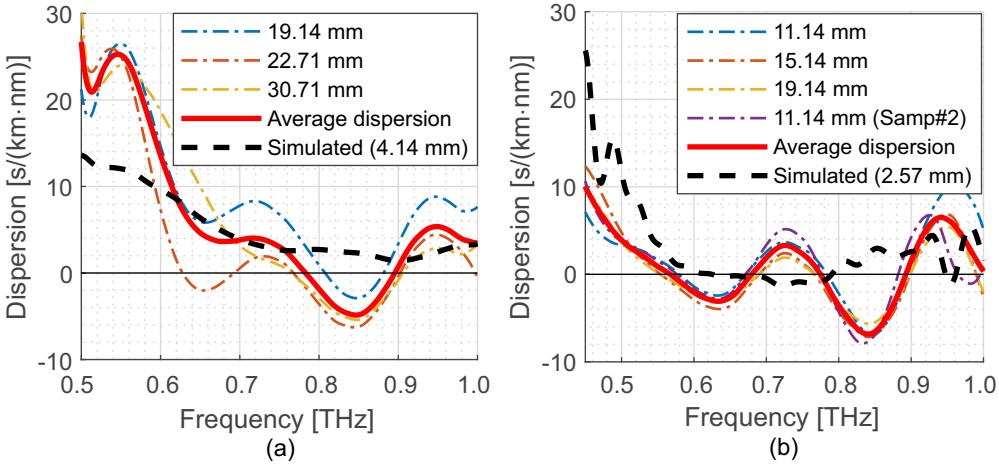

**Figure 3.** (**a**,**b**) show measured dispersion in waveguides of different $l_{WG}$ but same *ROC*, placed on quartz and HDPE substrate, respectively. Total and averaged dispersion of 3 waveguides with bend radius of 2 mm are compared with a simulated waveguide of same bend radius in (**a**). All waveguides shown in (**b**), along with the simulated one have a bend radius of 1 mm. The dispersion values are averaged over 30 GHz in order to reduce the influence of standing waves.

*4.2. Bends*

Bends are essential waveguide structures, that are also an integral part of other passive components, such as splitters. Bend losses are caused by (i) radiative loss at lower frequencies, which is akin to substrate leakage, and (ii) mode conversion losses at higher frequencies. For case (i), the losses by a bend can be described by the conformal mapping technique that maps the bent structure with constant refractive index to a straight waveguide with spatially dependent refractive index [47], including the surrounding media. Conformal mapping is detailed in the Supplemental Materials. The main result is summarized in Figure 4a. The conformal-transformed refractive index of media increases exponentially outwards from the centre of the circular bend as shown in the inset of Figure 4a. The lower the frequency, the smaller the effective refractive index of the guided mode and the larger the evanescent field. At a certain distance away from the waveguide, the conformally mapped refractive index becomes larger than the effective refractive index of the guided mode, causing radiative losses as illustrated in Figure 4a for 0.6 THz. At higher frequencies,

the mode concentrates more in the high index HRFZ-Si waveguide, resulting in larger effective refractive index of the guided mode and less extension of the evanescent fields. Consequently, the radiative losses decrease with increasing frequency [48] as demonstrated by the simulations for 0.8 and 1 THz.

For case (ii), mode conversion occurs as the power distribution of the guided mode increasingly shifts away from the waveguide centre with increasing frequency, as visible in the simulated power distribution for 1 THz in Figure 4a. This leads to mode mismatch at the transitions between straight and bent sections of the waveguide, which eventually leads to excitation of higher order modes at higher frequencies. These transition losses are inversely proportional to the square of the radius of curvature, $ROC^2$ [49]. Bends with decreasing $ROC$ at intersections to straight waveguide sections [50] or bend-offsets [51] reduce these mode conversion losses while keeping the effective bend radius, i.e., the space required for a low-loss bend, almost unaffected.

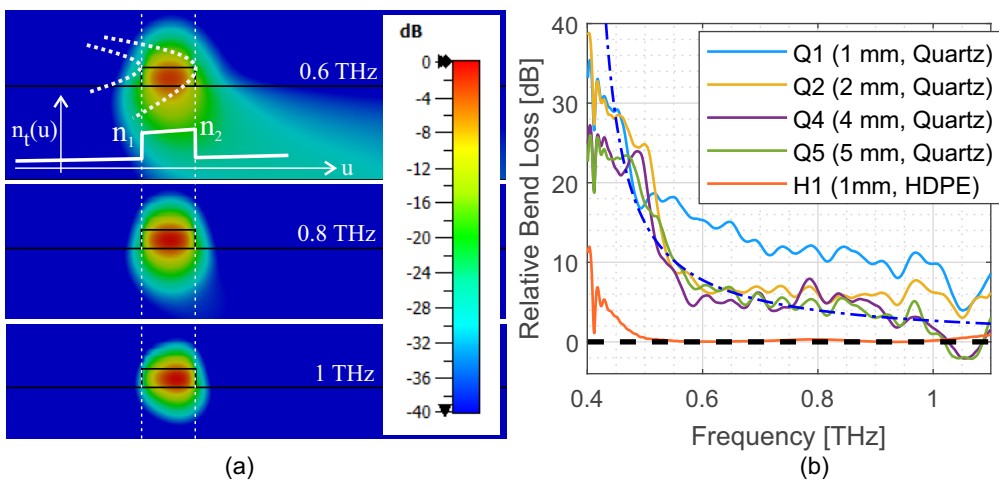

(a)　　　　　　　　　　　　　　　　　　　(b)

**Figure 4.** (**a**) Radiative losses and mode transformation at waveguide bends. The conformal-transformed refractive index $n_t(u)$ is plotted of a waveguide bend, with $ROC = 2$ mm where $n_1 = 3.25$ and $n_2 = 3.59$. (**b**) Measured losses for different curved waveguides on quartz normalized to a straight waveguide ($l_{WG} = 19.14$ mm) on HDPE. The apparent dips for Q1-5 at 1.05 THz is a measurement artefact arising due to standing waves. The blue dashed line is added to serve as a guide for the eye for Q4.

In this work, we characterize waveguides with circular bends only in order to determine the minimum bend radius. Scattering losses, induced by surface roughness of the waveguides ($<0.01\lambda_{THz}$ digs/sleeks), are much smaller compared to the aforementioned propagation losses and are not considered in the analysis. We examine the total loss for three different waveguides of length $l_{WG} = 19.14$ mm with bend radii of 1 (Q1), 2 (Q2), and 4 mm (Q4) on quartz. As the waveguides share the same length and transitions to free space, losses caused by the bends can be directly evaluated by comparing results from waveguides with different bend radii (Explicit calculation of bend losses per radian or per 90° turn requires a comparison of waveguides with a single bend and multiple bends. Due to the lack of structures with multiple bends, we opted for comparing structures with one bend each but different radius of curvature.). To remove losses caused by the straight sections and the coupling transitions from the waveguides to free space, the total loss data are normalized to the data of straight waveguide on HDPE substrate of same length (The straight waveguide data are calculated from a measured waveguide on HDPE, corrected by the simulated reduction in transmission efficiency between a straight and bent waveguide of same length.), as depicted in Figure 4b). Additionally, a longer waveguide (Q5, $l_{WG} = 22.71$ mm) with larger $ROC = 5$ mm is also plotted. We remark that below 0.5 THz the data quality for the waveguides on quartz suffers from increased loss by in- and out-coupling transitions.

Among the waveguides with quartz substrate, we observe that the waveguide with $ROC = 1$ mm (Q1) shows 3–5 dB higher loss than the rest over the whole frequency range, whereas waveguides with $ROC \geq 2$ mm have similar losses between 0.55–0.9 THz indicating that the wave loses negligible power through radiation at the bend. At $f_{\mathrm{THz}} > 0.9$ THz, we see losses for the waveguide with $ROC = 2$ mm (Q2), increasing with frequency with respect to waveguides with larger bend radii, which indicates the onset of mode conversion losses in Q2. The early onset of these losses compared to the simulations can be explained by the non-idealities in Q2, such as roughness of the side walls (c.f. Figure 1b)) and presence of adhesives between HRFZ-Si waveguide and quartz substrate, along with presence standing waves. The 5 mm $ROC$ bends, as well as waveguides with larger $ROC$ up to 10 mm (not depicted) show similar losses as Q4, proving that bend losses are negligible at radii $\geq 4$ mm up to at least 1.05 THz and according to simulations even beyond 1.6 THz. Larger $ROCs$ will further extend the higher frequency limit of the waveguides. In addition, Figure 4b) also depicts a bend with $ROC = 1$ mm (H1) on a HDPE substrate. Two dominant features are visible: first, the lower cut-off frequency shifts to around 0.45 THz despite having four times lower $ROC$. Similar to the straight sections, the lower refractive index of HDPE reduces radiative losses, both towards the substrate as well as at bends. Second, the observed loss over the whole frequency range is reduced by 5–7 dB. The lower transmission losses are partly due to the reduced usage of superglue between the waveguide and HDPE substrate. Additionally, the fundamental mode has a larger mode field diameter for H1, which also improves the in- and out-coupling efficiency at the intersections to free space.

## 5. Transitions to Active Devices

A key challenge is the realization of low loss transitions to sources and receivers as well as to other circuits such as electronic integrated circuits. The HDPE substrate already considerably improves the coupling efficiency. Still, the coupling loss at two positions (in- and out-coupling) totals to 18–26 dB over the whole frequency range from 0.4–1.05 THz. While for narrow band systems (bandwidths <50% of the centre frequency), the waveguide parameters do not change considerably allowing for close to perfect impedance-matching and mode field diameter matching with losses <1 dB [52], it is challenging to achieve matching for systems with a frequency coverage of several octaves. For instance, the mode field diameter becomes comparable to the size of the HRFZ-Si waveguide in the high frequency limit but extends considerably beyond at the low frequency end (c.f. Figure 4a), and, as a consequence, the wave velocity also shows strong frequency dependence. As opposed to narrowband circuits, a broadband topology cannot use any resonant elements, leading to necessary compromises between acceptable loss and achievable bandwidth.

An example of a broadband concept is the dielectric rod waveguide (DRW) antenna, basically a rectangular waveguide with a sharpened tip with a very small taper angle [53]. A DRW antenna is in principle compatible with the proposed topology; however, the optimum coupling position is again strongly frequency-dependent: the projection to the far field shows that high frequencies are emitted at the tip while low frequencies are emitted rather close to the base of the taper. This complicates in-coupling with fixed free space optics. For waveguides without transitions, the coupling losses using free space optics are high: For the presented waveguides, the vertical dimension of the waveguide extends only 50 μm with little evanescent field outside the HRFZ-Si at 1 THz (see Figure 4a, bottom). The Gaussian spot diameter of the free-space THz beam at the foci of TPX lenses is ≈1300 μm at 0.5 THz and ≈1000 μm at 1 THz. Thus, the waveguide cross-section is more than an order of magnitude smaller than the smallest beam diameter in the setup. To improve the coupling in horizontal direction to increase the coupling efficiency, waveguides are tapered-out at the input and output apertures to $1000 \times 50$ μm². With the given technology, however, the vertical axis cannot be tapered, causing the previously mentioned total loss of 18–26 dB for two transitions, even for the waveguide on HDPE and 22–28 dB for the waveguides on quartz. Such large losses are not acceptable for high dynamic range systems.

A detailed description of free-space coupling losses is given in section Section 4 of the supplemental material.

To circumvent this issue, we implement a near field coupling scheme at the receiver using planar end-fire Vivaldi antennas (VAs). Additionally, using this scheme eliminates the necessity of using optics such as lenses, parabolic mirrors or bulky silicon lenses, currently used in table-top THz sources and receivers. Consequently, this eases integration and miniaturisation of THz setups essential for manufacturing broadband THz photonic integrated circuits (PICs). Figure 5 shows an exemplary design of such a semi-integrated system. Within the operation range, VAs are frequency-independent [54], which means their beam pattern and impedance theoretically remain unaltered with frequency. Hence, VAs are suitable for broadband operation and the operating frequency range can be controlled simply by scaling the antenna dimensions. The tapered antenna structure slowly transforms an in-coupled wave to a guided wave which then couples to the active photoconductive element represented by the finger-like structure of the ErAs:InGaAs photoconductive material with a width of $D2 = 10$ µm and a thickness of 1.62 µm grown on a semi-insulating InP substrate. The finger electrodes are 1.5 µm wide, with a spacing of 2 µm. Further details on the material and a theoretical description of the receiving process can be found elsewhere [55]. The shown VA is a modification of antennas proposed by Abdullah et al. [56], where instead of having a superstrate atop the antenna, we thin the InP substrate supporting the active ErAs:InGaAs device [40] down to $h = 32$ µm to increase the antenna directivity in end-fire direction ($\theta = 90°$, $\phi = 0$) and to reduce squinting into the high index InP substrate ($n_{InP} \approx 3.48$). A substrate thickness $h$ fulfilling $0.005\lambda_0 \leq (n_{InP} - 1) \times h \leq 0.03\lambda_0$ is desired for improved impedance matching between the antenna and the radiated THz wave at the point of radiation [57]. For $h = 32$ µm, this condition is not fulfilled over the whole wavelength range. At the shortest wavelength, $h \geq 0.11\lambda_{0,min}$, however, the resulting reduced directivity can be compensated by increasing the surface current at the inner edges of the antenna blade [58] using slotted and circular corrugation structures. Additionally, metal directors further enhance the antenna performance and directivity at $f_{THz} > 0.9$ THz. The Vivaldi antenna is mounted on HDPE and directly brought in contact with the HDPE-mounted HRFZ-Si waveguide as shown in Figure 5c. Figure 5d shows a simulation at 0.875 THz proving very little mode mismatch and thus excellent coupling efficiency. The vast majority of the incoming power from the waveguide becomes absorbed by the finger structure located at position $A$. An insignificant amount of power ($< -20$ dB), escapes the antenna and is absorbed in the contact pads.

The current designs are optimized with respect to the lower cut-off frequency of the waveguides ($\approx 0.5$ THz) and the estimated frequency coverage of the CW photomixing setup. Optimum dimensions of the VAs are obtained by CST simulations for an operational frequency range of 0.5–1.5 THz. The dimensions are $D1 = 112.5$ µm and $D2 = 10$ µm corresponding to the maximum and minimum desired half effective-wavelengths, respectively. The operational bandwidth can be easily increased by scaling $D1$ and $D2$ appropriately. The antenna consists of an $\approx 400$ nm thick layer of Gold, with a Titanium adhesion layer. The VAs further feature extended metallic contacts behind the active element to extract the rectified THz current as shown in Figure 5a. The manufactured device is orders of magnitude smaller than commercially available state-of-the-Art THz lens-integrated receivers and is comparable in size to a strand of human hair as shown in Figure 5b.

To characterize the performance of the Vivaldi antenna, the receiver side of the waveguide setup shown in Figure 2a is replaced by the Vivaldi antenna as illustrated in Figure 5c. As we have no access to decent photomixer source material, we could not fabricate a VA-coupled source. Therefore, the source side of the setup is kept unaltered. For impedance matching reasons, a HDPE-mounted waveguide is used in conjunction with the HDPE-mounted receiver VA, where the receiver InP substrate de facto becomes an extension of the waveguide with $n_{Si} = 3.417 \approx n_{InP} = 3.48$.

To estimate the performance of the VA transition, Figure 6a shows the following plots for comparison: The bottommost plot (violet) shows the measured transmission

losses through the waveguides using free space, lens-assisted coupling at both in- and out-coupling positions of a waveguide as illustrated in Figure 2a. The measured losses are about 22–26 dB over the whole operation band, in agreement with the previous findings. The middle (blue) line represents the setup with the Vivaldi-antenna receiver as shown in Figure 5c. Compared to the lens-coupled setup, the losses are 5–14 dB less between 0.4–1.05 THz with a peak improvement of 14 dB around 0.68 THz. The topmost plot (black) shows the simulated achievable coupling efficiency when VAs are used for both in- and out-coupling to the waveguides. The plot shows a minimum insertion loss of ≈2 dB at 0.74 THz per VA-waveguide interface and a 3-dB bandwidth between 0.55–0.99 THz (relative bandwidth = 57.1%). The simulated coupling losses are however higher than the measured values in free-space and VA setup at frequencies above 1.2 THz, which indicates the necessity for further optimization of the VAs at the upper end of the operational band. Furthermore, simulations indicate that when a VA is implemented on the source side, the waveguide should support only one mode in the vertical dimension in order to prevent excitation of higher order modes. This imposes another limit on the upper cut-off frequency of about 2 THz for a 50 μm tall waveguide.

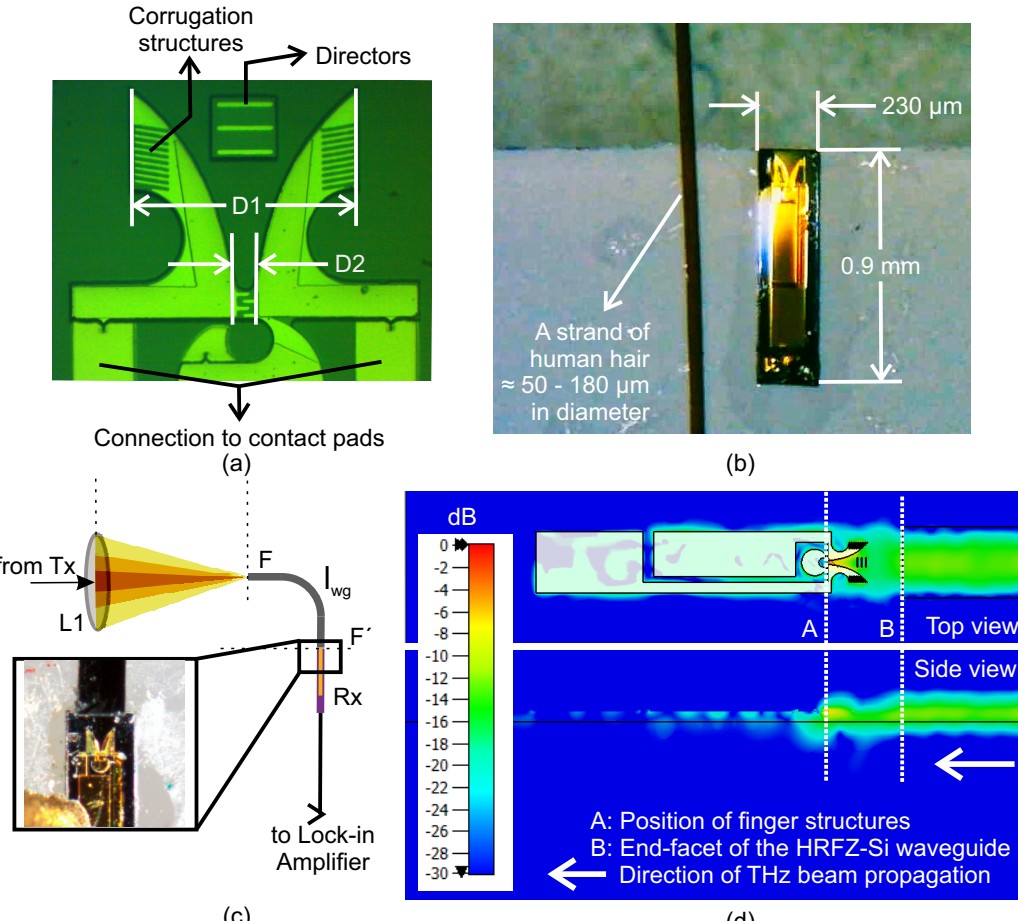

**Figure 5.** (**a**) Micrograph of a VA fabricated in-house. (**b**) Receiver integrated with a VA, which is ≈230 μm wide, 0.9 mm long and with a substrate thickness of only 32 μm mounted on HDPE. The substrate is thinner than a human hair. Schematic in (**c**) shows the experimental setup with Vivaldi antenna coupled THz receiver. An actual microscopic image of the coupled waveguide is in the inset. (**d**) shows simulation of the average power flowing from the waveguide into VA at 0.875 THz. The simulated coupling efficiency of the waveguide-VA interface at 0.875 THz is −4.3 dB.

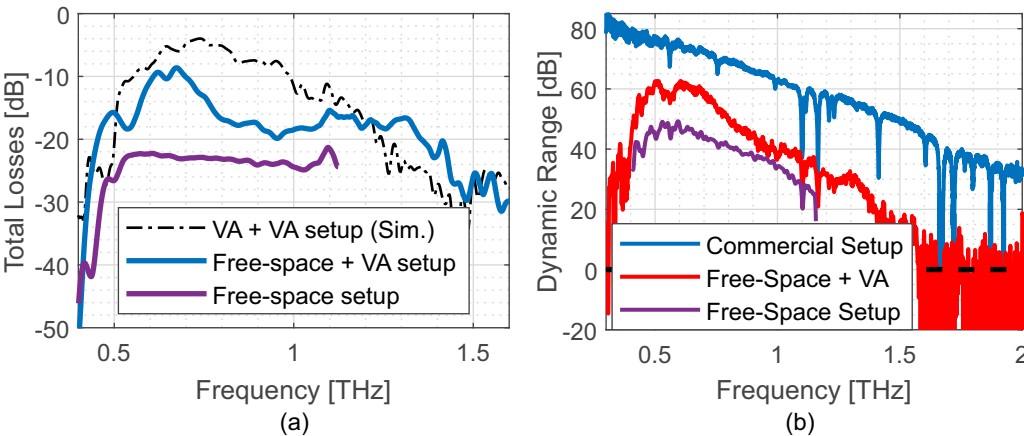

**Figure 6.** (**a**) shows the total losses of the semi-free-space measurement setup shown in Figure 5c. The measured losses are averaged over 25 GHz. (**b**) shows the dynamic range comparison between commercial setup, free-space setup using waveguides and semi-free-space setup with VA receiver.

In summary, the VA features an improvement of the coupling efficiency of 5–14 dB above 0.4 THz per coupling position. Using VA at one coupling position extends the operational range of the demonstrated system with a waveguide-coupled detector and a lens-coupled commercial p-i-n diode source to 1.6 THz as shown in Figure 6b. We expect an increase of the operational bandwidth of the system to at least 2 THz if near-field coupled devices are implemented at both the source and receiver side. The VA also shows a less pronounced roll-off than logarithmic-periodic or spiral antennas as the radiation resistance is smaller. Furthermore, there is no requirement for a silicon lens, simplifying and compacting the setup. Experimental findings also show reduced reflections by the VA antenna due to improved mode field diameter- and impedance matching, reducing the amplitude of undesired standing waves in the waveguide. By replacing the active element by a short transmission line with enforced end contacts a GS coupler to a metal-based electronic integrated circuit can be realised. This may enable future on-chip measurement systems such as photonic vector network analysers with operational bandwidths as demonstrated here.

## 6. Couplers

Power splitters and combiners are essential components for photonic circuits. Figure 7a shows a 3-dB coupler/splitter consisting of two symmetrically designed output arms, with sinusoidal and circular bends with 1 mm bend radii. The structure is fabricated on a quartz substrate. The splitter is characterized in the free-space setup (c.f. Figure 2a) between 0.45–1.05 THz. It shows an equal splitting ratio at a power level of 4.8–6 dB in its two arms, which is in excellent agreement with the simulation. Since, the coupler was designed with $ROC < 4$ mm, higher order modes are excited at frequencies <0.9 THz resulting in excess losses and an undulating transmission characteristics. The performance can thus be further improved by increasing the bend radii. The simulated isolation between the two output arms is >40 dB for frequencies >0.65 THz, and reaches a minimum of 32 dB at 0.582 THz.

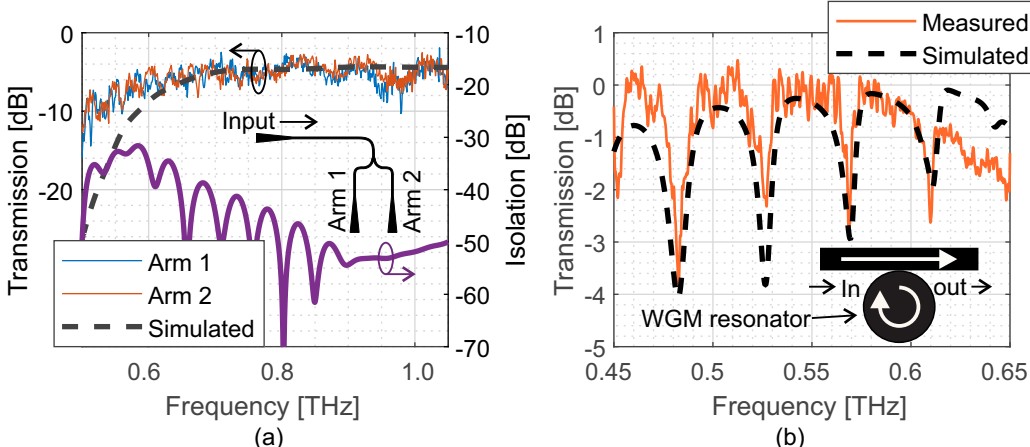

**Figure 7.** (**a**) Measured and simulated transmission coefficient for a 3 dB splitter. Plot in violet shows the simulated isolation between two output arms. (**b**) Transmission characteristics of a filter based on a WGM resonator. The white arrow in the WGM resonator schematic denotes the propagating wave through the waveguide.

## 7. Frequency-Selective Filter Example

A whispering gallery mode (WGM) resonator is measured between 0.45–0.65 THz with a VA receiver. The resonator structure is a 50 µm high silicon disk, matching the height of the waveguide, with a design radius of 400 µm. It is placed in direct contact with the straight section of a waveguide. Figure 7b shows the transmission coefficient, normalized to a reference measurement of the bare waveguide without the resonator. Resonance peaks in the measured and simulated data match excellently; however to a simulated resonator radius of 408 µm. The resonator has a free spectral range of around 45 GHz and the steepest resonance peak at 0.482 THz shows a quality factor of 38.9. The simulated quality factor is slightly lower than the measured value. These may have several causes, such as a slight misalignment (i.e., slight distance between WGM and waveguide), non-straight sidewall of the WGM, causing a small gap at the coupling position, or simply non-zero losses of all involved materials. We remark that for all modes, the coupling was under-critical as critical coupling cannot be achieved with the given waveguide width and considerably small amount of evanescent field in the horizontal direction. Resonators incorporated within the waveguides or non-planar WGM resonators [59] can serve as better alternatives. Attaching a second waveguide to the WGM results in an add drop filter for specific frequencies [60]. Multiple resonators with different mode spectra allow engineering wavelength division multiplexers [61].

## 8. Conclusions

We demonstrated broadband characterization of a dielectric waveguide architecture based on HRFZ-Si waveguides on crystalline quartz and HDPE substrates between 0.4–1.05 THz in a free-space setup. Waveguides on quartz substrates show a very low attenuation constant below 0.4 dB/cm between 0.65–0.9 THz. At higher frequencies, standing waves introduce measurement errors that only allow to provide an upper bound of about 1.7 dB/cm. The minimum bend radius for low loss circular curved sections is found to be larger than 2 mm, whereas 4 mm bends already show close to ideal performance. HDPE substrates reduce leakage and coupling losses as compared to quartz and decrease the lower cut-off frequency to ≈0.41 THz. The challenge of low loss transitions to sources, receivers and planar metallic waveguide-based components in general was addressed with end-fire Vivaldi antennas where we have demonstrated a reduction of the coupling losses compared to a free space transition by 5–14 dB per coupling position. Furthermore, the reduced losses increase the operation range of the waveguide-based homodyne THz system consisting of a waveguide-coupled VA detector and a commercial lens-coupled p-i-n diode source, from 0.5–1.6 THz, which is the largest reported operational bandwidth

of such an integrated THz setup up until now. Further characterized components include a 3-dB splitter and a filter based on a whispering gallery mode resonator. The proposed waveguides and waveguide components will significantly facilitate development of miniaturized table-top broadband THz systems, such as photonic THz-VNAs and photonic spectrum analysers.

**Supplementary Materials:** The following are available at https://www.mdpi.com/article/10.3390/photonics8110492/s1: 1. Details of the experimental setup, 2. Conformal Transformation, 3. Calculation of waveguide dimensions, 4. Simulation results, 5. Standing waves and higher order modes and 6. On-top WGM Resonators.

**Author Contributions:** Conceptualization, A.K.M. and S.P.; methodology, A.K.M. and S.P.; software, A.K.M. and M.X.; validation, A.K.M., M.X. and S.P.; formal analysis, A.K.M.; investigation, A.K.M. and M.X.; data curation, A.K.M. and M.X.; writing—original draft preparation, A.K.M.; writing—review and editing, S.P.; visualization, A.K.M.; supervision, S.P.; project administration, S.P.; funding acquisition, S.P. All authors have read and agreed to the published version of the manuscript.

**Funding:** The research was funded by European Research Council (ERC) starting grant Pho-T-Lyze, grant agreement number 713780.

**Data Availability Statement:** All the data supporting the findings are available from the corresponding authors upon request.

**Acknowledgments:** We acknowledge Irina Harder and the group for Nanofabrication TDSU1 from the Max Planck Institute for the Science of Light (MPL), Erlangen for their expertise on the DRIE process, along with providing access to the MPL cleanroom and dry etching facilities, and 3D-Mircomac AG, Chemnitz for micro-machining HRFZ Si waveguides. We also acknowledge support from the Deutsche Forschungsgemeinschaft (DFG: German Research Foundation) and the Open Access Publishing Fund of Technical University of Darmstadt for the article processing charges.

**Conflicts of Interest:** The authors declare no conflict of interest.

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
