# Peer review of "Broadband Terahertz Photonic Integrated Circuit with Integrated Active Photonic Devices"

_photonics, doi:10.3390/photonics8110492_

Round 1

Reviewer 1 Report

Despite the fact that the topic of dielectric waveguides is a well-studied area, the authors try to present a systematic approach to the study of it and demonstrate interesting results in the wide frequency range. The summaries are almost in all experimental sections that make the manuscript clear for understanding. The manuscript conclusion could be useful in choosing between quartz or polymer substrate for future integrated photonic devices. Unfortunately, the manuscript just a bit touches the waveguides' integration with photonic devices, as a coupler/splitter and WGM resonator. These results are important steps in promising integration of electronics and photonic devices for the terahertz frequency range.

 The overall quality of the presented results is good and sufficient for publication in MDPI Photonics.

However, before being accepted for publication, the following points should be addressed:
- Explain the photomixing technique for the generation of THz signal.
Because in the manuscript it looks referred to frequency-domain and in supplementary material data are presented in time-domain (Fig S4);
- The description of coupling losses (Supp. mat. Sec. 4) could be particularly implemented in the manuscript. It gives more clear part; 

Minor remarks:
- Label Rx_{LP} (line 162) does not use further;
- Acronyms ICP  is used one time;
- to do the same letter case for L_{WG} in the text and in figures (Fig.2a and S1b and c.);
Supplementary materials:
- Fig.S3.a difference description in the label W2 and in the caption W1.
- line 126, there is a missed number of references.

Reviewer 2 Report

This manuscript presents a broadband terahertz photonic integrated circuit with integrated active photonic devices. A high resistivity silicon-on-insulator-based multimodal waveguide topology including active components as well as passive components (couplers/splitters, bends, resonators) with investigated over a frequency range of 0.5 − 1.6 THz. From my point of view, the manuscript is clearly written, with experimental results to support the performances of device. It could be useful for miniaturized table-top broadband THz systems. The paper can be published after the authors considered a few points:

  1. Proper referencing is missing in sections 1. There are other related approaches of high resistivity silicon terahertz photonic integrated circuit and resonators. e.g., J. Xie et al., " Terahertz integrated device: high-Q silicon dielectric resonators", Optical Materials Express 8 (1), 50-58 and Z. Wang, et al., " Voltage-actuated thermally tunable on-chip terahertz filters based on a whispering gallery mode resonator.", Optics Letters 2019, 44, 4670-4673. These have been summarized in A Review on Terahertz Technologies Accelerated by Silicon Photonics, Nanomaterials 11 (7), 1646.

  1. In page Fig3a) and b), the authors state that the standing waves cannot originate from mode beating, then So what causes standing waves? Could the authors give more detailed explanation?

  1. It will be better if the authors give an accurate coupling efficiency in Fig. 5d).

  1. In Fig. 6a), why are the experimental coupling losses better than the theoretical simulation values in some frequency range when using the same device structure?

  1. Kindly add detail performances of the resonator, e.g. Q-factor and free spectral range (FSR).

  1. At the beginning of the paper, the multiple mode property of waveguide was analyzed, but the high-order mode was not directly verified in the subsequent waveguide transmission and whispering gallery mode (WGM) resonator experiments. The power flowing simulation in Fig. 5d) was still the fundamental mode, and the resonance spectra in Fig. 7b) and S5b) seemed to see only one mode resonance (which may also be the fundamental mode resonance). It could be better to indicate and analyze the actual mode in the experiment to reveal the performance of the proposed multiple mode waveguide.

Reviewer 3 Report

Dielectric waveguides are designed for the terahertz range, using a silicon-on-insulator platform. The propagation characteristics of the waveguides are explored experimentally, and then the waveguides are coupled to an InP photomixer detector that uses a vivaldi antenna. The authors also show a few different passive devices. 

This work presents some novelties and new achievements that are worthy of publication in MDPI’s  Photonics journal, but there are a few issues with the processing and interpretation of results that must be addressed. 

  1. The authors use the term “cut-on” frequency. The term should always be “cut-off,” even if we are discussing it from the viewpoint of a frequency where waves begin to propagate. 

  1. Photomixer sources have previously been coupled to silicon terahertz waveguides. The authors should cite this paper (https://doi.org/10.1109/TAP.2014.2387419), as it is highly relevant to this manuscript.

  1. The single-mode frequency range of the dielectric waveguides is from 0.5 THz to 0.75 THz. This information should be given prominently in the abstract and conclusion of the manuscript, for honest and transparent disclosure. Related to that point, the authors state that “The waveguides offer lower loss than commonly implemented hollow metal waveguides above 1 THz,” but this comparison is not entirely fair, because it omits the fact that their dielectric waveguide is multi-mode in this range, whereas the hollow waveguide is not. 

  1. The authors give the dielectric constant of intrinsic silicon as ε=11.62 ± 0.05, but I believe the true value is closer to ε=11.7 (https://doi.org/10.1364/JOSAB.21.001379). 

  1. Why is the word “characterize” italicized on line 86? 

  1. On line 91, the authors say that HDPE and SiO2 “...are known as two of the lowest loss materials up to ∼ 2 THz or even higher.” Isn’t the loss of silicon lower? Or do the authors specifically mean something like “materials for which index is sufficiently low to be used as support, with an appreciable leakage-free bandwidth”? In which case, the authors should make this explicit. Also, authors should be be aware of COC, which is not only lower-loss than HDPE in the terahertz range, but I believe it is also lower-index (i.e. to help with reducing leakage). 

  1. On the line 114 the authors say that “A polyethylene substrate reduces the lower 3-dB cut-off frequency to 0.45 THz for a waveguide of length 19.14 mm.” This is a very unusual way to report waveguide cut-off. Firstly, length should not have any relevance to it, as cutoff is just the lowest frequency for which the mode is non-leaky. So the 3-dB limit of a finite-length waveguide has nothing to do with it. Instead, please talk about the frequency at which the mode starts to leak into the substrate. 

  1. The authors have stated on line 120: “...an upper frequency limit to the single mode behaviour is imposed by passive elements such as bends, splitters etc., which, if not properly designed, will lead to mode conversions to higher order or even radiative modes.” There are several problems with this. Firstly, if higher-order-modes are undesired then the upper-frequency limit is set by the cut-off of those higher modes. As such, it is a feature of the waveguide itself, and not of passive devices. Secondly, the authors do not really specify what they mean by a “properly designed” device. The passive devices that they give in later sections appear quite standard and conventional, with no special technique or effort to suppress undesired modes. 

  1. Related to the previous point, the authors say on line 125 that “We show in the upcoming sections that with appropriate bend designs and controlled excitation, the dielectric waveguides can be practically operated in its fundamental mode...”. However, they do not provide strong evidence that higher-order modes have been avoided. They try to argue based on simulated time-domain traces, but this is quite a strange choice of analysis, since EM simulation packages will allow users to inspect the power delivered to different modes directly. For this reason, this argument seems a little bit deceptive. 

  1. Related to the previous two points, the authors say on line 183 that “Above 0.9 THz, standing waves in the waveguides affect the data quality.” But this is basically the same frequency where the higher-modes start to propagate. Clearly these effects are due to higher-order modes. 

  1. The authors say on line 208 that “...the standing waves are present over the whole investigated frequency range and therefore cannot originate from mode beating as higher order modes cannot propagate at the lower end of the investigated frequency range.” This argument is not very convincing. Granted, higher-order modes cannot propagate leakage-free below 0.75 THz, but they can still propagate as leaky-modes, and this can produce the same type of ripple in the frequency domain. Also, there could be a combination of both standing waves and higher-order modes.

  1. Bending loss is evaluated in an unusual way. Can’t you normalize against a straight waveguide, or compare the loss of a waveguide with multiple bends against one with only one bend? 

  1. In the conclusion, the authors write “Further, the reduced losses increase the operation range of the waveguide-based THz system from 0.5 − 1.6 THz, which is the largest reported operational bandwidth of such an integrated THz setup up till now.” To call the measured range of 0.5-1.6 THz the “operation bandwidth” is not quite correct, given that the single-mode region of this system is more like 0.5-0.75 THz. 

Round 2

Reviewer 3 Report

The authors have responded to all of the issues that I have raised, but further revisions will be necessary.  

1. Ok

2. Ok

3. Put the single-mode bandwidth of your waveguide in the abstract. 

4. Ok

5. Ok

6. Ok 

7. In general, waveguide cut-off has nothing to do with length. They are unrelated. Are the authors able to provide reference for other work in the photonics literature that uses this way to report the cut-off?

The field should not leak if it is above the light-line. This is fundamental. The fields indicated by the authors could be a simulation artifact related to the injection of terahertz waves at the input port. 

8-11. I think there is a lot of confusion about how to determine the existence of higher-order modes. The authors have made a lot of arguments to say that they have suppressed higher-order modes, but none of them are totally ironclad. So, rather than argue every single response individually, I will give instructions to clarify the question of higher-order modes. 

The authors appear to have access to simulation software. This makes it possible to clearly analyse the amount of each mode that is produced. Just simulate all of the devices again, and include S-parameters for all propagating modes in the supporting information for this paper. Then, rather than just giving several qualitative statements about the suppression of undesired modes, you can just refer to the plots, and give precise, quantitative information. That way, readers can make up their own minds whether the level of these modes is acceptable or not. This is the most clear and transparent approach to report a critical aspect of this work. 

12. Why didn’t you fabricate enough devices to support rigorous characterization of bending loss? Please include a note about this in the manuscript text. 

13. The authors’ definition of the device’s operation bandwidth as “the spectral coverage of a system where it works properly and remains functional” is based upon conventions of THz-TDS systems. But the device that they report is just a photomixer-based detector, and not a whole THz-TDS system. So 3-dB bandwidth is actually more appropriate. 

Consider: If you used a more powerful source for this experiment, and this pulls more spectrum out of the noise floor, then would this mean you have increased the detector’s bandwidth? Of course this is nonsense because it is the same detector. 

At points where authors report bandwidth of “0.5-1.6 THz,” they should be very explicit about how they have defined this bandwidth. A waveguide-coupled terahertz detector with an operation bandwidth of 3:1 would be very impressive, but that is not what the authors have achieved. 

Round 3

Reviewer 3 Report

Ok it is pretty much fine now.